# Role of Piezo1 in Terminal Density Reversal of Red Blood Cells

**DOI:** 10.3390/cells13161363

**Published:** 2024-08-16

**Authors:** Kuntal Dey, Ankie M. van Cromvoirt, Inga Hegemann, Jeroen S. Goede, Anna Bogdanova

**Affiliations:** 1Red Blood Cell Research Group, Institute of Veterinary Physiology, Vetsuisse Faculty, University of Zurich, CH-8057 Zurich, Switzerland; kuntaldey.edu@gmail.com (K.D.); ankievancromvoirt@hotmail.com (A.M.v.C.); 2Department of Medical Oncology and Hematology, University Hospital and University of Zurich, CH-8091 Zurich, Switzerland; ingahegemann@gmx.de; 3Department of Hematology, Kantonsspital Winterthur, CH-8401 Winterthur, Switzerland; Jeroen.Goede@ksw.ch; 4Zurich Center for Integrative Human Physiology, University of Zurich, CH-8057 Zurich, Switzerland

**Keywords:** red blood cells, senescence, calcium, terminal density reversal, Piezo1

## Abstract

Density reversal of senescent red blood cells has been known for a long time, yet the identity of the candidate ion transporter(s) causing the senescent cells to swell is still elusive. While performing fractionation of RBCs from healthy individuals in Percoll density gradient and characterization of the separated fractions, we identified a subpopulation of cells in low-density fraction (1.02% ± 0.47) showing signs of senescence such as loss of membrane surface area associated with a reduction in band 3 protein abundance, and Phosphatidylserine (PS) exposure to the outer membrane. In addition, we found that these cells are overloaded with Na^+^ and Ca^2+^. Using a combination of blockers and activators of ion pumps and channels, we revealed reduced activity of Plasma membrane Ca^2+^ ATPase and an increase in Ca^2+^ and Na^+^ leaks through ion channels in senescent-like cells. Our data revealed that Ca^2+^ overload in these cells is a result of reduced PMCA activity and facilitated Ca^2+^ uptake via a hyperactive Piezo1 channel. However, we could not exclude the contribution of other Ca^2+^-permeable ion channels in this scenario. In addition, we found, as a universal mechanism, that an increase in intracellular Ca^2+^ reduced the initially high selectivity of Piezo1 channel for Ca^2+^ and allowed higher Na^+^ uptake, Na^+^ accumulation, and swelling.

## 1. Introduction

The average lifetime of mature red blood cells (RBCs) is around 120 days. During this time of RBCs’ aging, loss of deformability, ionic imbalance, and irreversible damage of proteins become prominent. As a result, most of the senescent cells become dense and rigid and are ultimately cleared from circulation [1,2,3,4,5]. It is well established that an aberration in the function of ion channels and pumps drives RBC dehydration associated with aging [5]. Calcium ions (Ca^2+^) play crucial roles in the physiology of RBCs such as controlling cellular ion and water content, maintaining intracellular redox state, proteolytic activity of µ-calpain, and thereby regulating the aging process of these cells [6,7]. The dynamic equilibrium of intracellular free Ca^2+^ ions, [Ca^2+^]_i_, is maintained by Ca^2+^-permeable ion channels and the Ca^2+^ pump results in a steady-state transmembrane ionic gradient in the order of 50,000-fold. While Ca^2+^ influx in human RBCs is mediated by several types of nonselective cation channels and putative leaks [8], Plasma membrane Ca^2+^ ATPase (PMCA) is the sole transporter mediating Ca^2+^ efflux [9]. The inability to sustain the basal cytosolic free Ca^2+^ concentration of about 30–60 nM is a nonspecific marker of RBC distress reported for a number of hereditary hemolytic anemias and dialysis patients, and in inflammatory states [10,11].

While studying time-dependent changes in cellular hydration in K^+^ permeabilized RBCs, Bookchin et al. [12] found a small fraction of cells (0.03 to 4%) in healthy donors or sickle cell patients showing senescent features that fail to dehydrate, in contrast to the majority of the senescent cells which become progressively dehydrated by the net loss of KCl and water. The inability to maintain transmembrane Na^+^/K^+^ gradients in these cells was associated with an increase in the levels of glycated hemoglobin (HbA1c), high passive permeability to Na^+^ and Ca^2+^ mediated by the nonselective cation conductive pathway of unknown molecular identity (P_cat_) [13]. These studies indicated the presence of a small fraction of cells that swell while becoming physiologically senescent, showing “reversal in their densities”. To describe this phenomenon, the authors Bookchin and Lew et al. first coined the term “Terminal Density Reversal” (TDR) [13]. The low abundance of such swollen senescent cells was explained by their instantaneous removal or by the heterogeneity in the population of senescent cells, only a few of which may reverse their density. Activation of the nonselective cation current P_cat_ was considered one of the mediators of this density reversal. It was also suggested by Lew et al. [13] that the enhanced Ca^2+^ content in these senescent cells drives the activation of P_cat_ leading to enhanced water and Na^+^ accumulation, which in turn results in the lowering of cell density. Nevertheless, the molecular identity of P_cat_ remains elusive.

While revisiting the senescence markers for the aging of RBCs in healthy individuals, we observed a certain percentage of senescent-like cells with low density [14]. In this current study, we characterized the occurrence of aging/senescent markers and explored the [Ca^2+^]_i_ and intracellular [Na^+^] ([Na^+^]_i_) ionic contents of these individual cells aiming to identify possible candidate protein(s) responsible for the density reversal of these cells.

## 2. Materials and Methods

### 2.1. Blood Samples

Lithium-heparin-preserved mixed venous blood samples from healthy donors (with informed consent), were used in this study. The study was approved by the ethics committees of the University of Heidelberg, Germany (S-066/2018), and the University of Bern, Switzerland (2018-01766), or harvested for the calibration of devices in the hematological labs of Cantonal Hospital Winterthur and University Hospital Zurich. 

### 2.2. Red Blood Cell Density Separation

Isotonic Percoll density gradient was used to fractionate RBCs according to their density and enrich fractions with young, mature, or senescent RBCs using fractionation in 90% isotonic Percoll self-forming gradient prepared on plasma-like medium with or without 1.8–2.0 mM Ca^2+^ as stated elsewhere [15]. Low- (L), medium- (M), and high- (H) density fractions were collected, washed three times, and resuspended in the plasma-like medium for further characterization.

### 2.3. Hydrolytic Activity of PMCA

RBCs’ PMCA hydrolytic activity was assayed as described by Zemlyanskikh et al. [16] with several modifications. Aliquots of washed RBCs (final hematocrit 10–15%) were added to the medium of the Ca^2+^-free buffer supplemented with 0.015% Saponin. The cells were incubated for 20 min on ice. The mixture was transferred to 37 °C initially for 5 min and CaCl_2_ was supplemented to reach the final free Ca^2+^ concentration of 10 µM. To initiate the hydrolytic PMCA activity, ATP (3 mM final concentration) was added and the reaction was allowed to occur for 10 min, the mixture was transferred on ice, and then stopped by ice-cold 10% TCA. After centrifugation, the released Pi was determined according to Dey et al. [17].

### 2.4. Flow Cytometry

The following parameters were detected using flow cytometry: the geometric mean of forward (FS) and side (SS) scatter, immature reticulocyte counts (CD71 staining), abundance of PS on the outer membrane (Annexin V staining), RBC membrane surface area for band 3 (eosin 5-maleimide, EMA staining), [Ca^2+^]_i_ (Fluo-4 AM), and [Na^+^]_i_ (CoroNa™ Green-AM).

Triple staining was performed with CD71 (Anti Human CD71, eBioscience, San Diego, CA, USA), Annexin V (Invitrogen, Waltham, MA, USA), and Fluo-4 AM (Invitrogen, 2 µM) to distinguish between the young (CD71+) and senescent cells (Annexin V+) and their [Ca^2+^]_i_ content. [Na^+^]_i_ was measured by utilizing the dye CoroNa™ Green-AM by following the protocol of Iamshanova et al. [18] as well as Negulescu et al. [19] with substantial modifications: (i) the dye loading was performed at 37 °C for 45–60 min; (ii) for calibration of CoroNa Green-AM, fluorescence intensity was recorded for the clamped cells with fixed intracellular Na^+^ concentrations in the plasma-like buffer. To study the effects of different modulators on [Na^+^]_i_, the Piezo1 channel was activated by 1.5 µM Yoda1 or blocked by 2.5 µM GsMTx-4; Na^+^, K^+^-ATPase (NKA), and PMCA activities were blocked with 100 µM Ouabain or 1 mM Na-orthovanadate. Additionally, in flow cytometry, we examined the capacity of the cells to swell in response to hypoosmotic stress [20]. 

### 2.5. Statistical Analysis 

Statistical analysis was performed in GraphPad (GraphPad Prism version 8 (8.4.3)). Afterward, a paired parametric (Student’s paired *t*-test) or paired non-parametric test (Wilcoxon matched-pairs signed rank test) was used to compare the different groups. 

## 3. Results

### 3.1. Low-Density Senescent-Like Cells

After isolation of fractions enriched with young (low density, L-fraction), mature (medium density, M-fraction), and senescent (high density, H-fraction) RBCs from the blood of healthy donors, we monitored Ca^2+^ levels in these fractions by flow cytometry. Each of the fractions contained a subpopulation of cells overloaded with Ca^2+^ (High Ca^2+^ fraction Gate A, Figure 1A). In L-fraction, the percentage of these high Ca^2+^ cells ranged from ≈0.5% to ≈ 2% of the total cell population. While most cells (Gate B in Figure 1A) did not differ in Ca^2+^-dependent fluorescence intensity between the fractions (Figure 1B), free Ca^2+^ content of the high Ca^2+^ fraction was maximal in the L-fraction compared to the RBCs in M- and H-fractions (Figure 1C). The amount of high Ca^2+^ cells in L-fraction exceeded that in the mature M-fraction (Figure 1D). We further co-stained the cells in the L-fraction with Annexin V for PS exposure to the outer leaflet of the membrane bilayer, anti-CD71 antibody for early reticulocytes, and with Fluo-4 for [Ca^2+^]_i_ levels. The majority of the Ca^2+^-overloaded cells were Annexin V+ (more than 90% of Annexin V+ cells were in the high Ca^2+^ fraction) and vice versa, meaning the majority of Annexin V+ cells were high in [Ca^2+^]_i_, while CD71+ cells were not (Figure 1E,F). It is interesting to note that the addition or omission of Ca^2+^ during Percoll density gradient separation did not affect the number of Annexin V+ or CD71+ cells in L-fraction (Appendix A). Comparison of the band 3 protein abundance between Annexin V+ cells and CD71+ reticulocytes visualized as the Eosin Maleimide staining intensity revealed a significant loss of band 3 protein-associated membrane (Figure 1G) associated with PS exposure to the outer membrane leaflet. These features identified the Ca^2+^-overloaded Annexin V+ subpopulation of the L-fraction as damaged senescent cells, which will be addressed further on in the study as “Senescent-like High Ca^2+^ cells” (SLHC cells). One could expect that activation of the Gardos channels in RBCs caused by Ca^2+^ overload as seen in Annexin V+ cells should have resulted in K+ loss and dehydration of these cells. However, SLHC cells were overhydrated and had low density. In the next set of experiments, we have addressed the possible causes of their overhydration.

### 3.2. Intracellular Na^+^ in SLHC Cells 

Passive uptake of Na^+^ by RBCs following the transmembrane Na^+^ gradient is a common cause of cell swelling. Thus, we aimed to compare [Na^+^]_i_ levels in single RBCs forming SLHC and CD71+ subpopulations using flow cytometry. We tested CoroNa Green-AM fluorochrome for detection of the relative changes in [Na^+^]_i_. Initially, we identified the optimal concentration of the dye to detect [Na^+^]_i_ in RBCs, which appeared to be similar to the previously reported value of 10 µM. The sensitivity of CoroNa Green for detection of the differences in [Na^+^]_i_ content in the range from 10 to 100 mM in RBCs was then assessed (Appendix A). For testing the sensitivity of the dye to the changes in the [Na^+^]_i_, we clamped the erythrocytic Na^+^ concentrations to be within the range from 10 to 100 mM (the constant extracellular osmolarity was maintained by choline chloride) using a Na^+^ ionophore Gramicidine (20 µM). During the experiments, we observed the differences in Na^+^-independent signal intensity between the SLHC cells and the rest of the cells and, therefore, had to make separate calibration curves for SLHC cells as shown in Appendix A. In our subsequent data, additionally, we confirmed that high Ca^2+^ levels in SLHC cells did not impact the Na^+^-dependent CoroNa Green signal. For each quantitative experiment, a fresh batch of the dye was prepared, and calibration performed. We applied the optimized protocol to the non-fractionated RBCs as well as to the cells forming the L-fraction where the RBCs were co-stained with CoroNa Green, anti-CD71 antibodies, and Annexin V. By using the calibration curve (Appendix A), we found that [Na^+^]_i_ in unfractionated RBCs (designated as whole blood, WB) is ~9.9 ± 1.35 mM (*n* = 3). In CD71+ cells, CoroNa Green dye fluorescence intensity was non-significantly higher than whole blood (Figure 2A,B). Importantly, from Figure 2A,B, it is apparent that Annexin V+ cells (in other words SLHC cells) have very high [Na^+^]_i_ content, portraying that these cells largely lost their transmembrane Na^+^ gradient. Notably, we could not detect the impact of fractionation on Percoll density gradient on the CoroNa Green dye fluorescence intensity in both CD71+ and Annexin V+ cells.

We then tested the impact of the modulation of nonselective cation channel Piezo1 as well as of the function of NKA, the sole Na^+^ extruder in RBCs, on the [Na^+^]_i_ in SLHC cells. To do so, the cells from L-fraction were incubated in the absence or presence of extracellular Ca^2+^, NKA blocker Ouabain, and Piezo1 modulators Yoda1 and GsMTx4. For each condition, changes in [Na^+^]_i_ in CD71+ and SLHC cells were determined using CoroNa Green-AM. We did not find any significant changes in the Na^+^-dependent fluorescent signal in CD71+ cells upon modulation of the activity of Piezo1 channel and/or NKA (Figure 2C). On the other hand, a reduction in [Na^+^]_i_ was observed in the case of Annexin V+ cells treated with Yoda1 in Ca^2+^-containing medium. However, it was not observed anymore if the cells were additionally treated with Ouabain, suggesting that suppression of the passive Na+ uptake mediated by the Piezo1 channel allowed NKA to reduce [Na^+^]_i_ content (Figure 2C). In our next section, we explored the causes of the reduction in the [Na^+^]_i_ signal in SLHC cells. Inhibition of Piezo1 in the SLHC cells resuspended in Ca^2+^-containing medium resulted in the same inhibitory effect, although less pronounced than that in the presence of Yoda1 (Figure 2D). These data suggested that (i) Piezo1 may play an important role in Na^+^ overload in SLHC cells; and (ii) Piezo1 transports Na^+^ in a Ca^2+^ -dependent manner in SLHC cells. In our subsequent study, we examined the dynamics of Ca^2+^ movement through Piezo1 in these SLHC cells.

### 3.3. Ca^2+^ Dynamics in SLHC Cells

We proceeded with the investigation of the role of PMCA and Piezo1 in mediating Ca^2+^ overload in SLHC cells. As it was not possible to apply single-cell methodology, hydrolytic activity of PMCA was determined in all the cells forming L-, M-, and H-fractions (Figure 3A). The lowest hydrolytic activity of PMCA was detected in RBCs from the H-fraction that were on average older than the cells from all other fractions. We did not find any change in the abundance of the PMCA4b protein in different fractions by Western blotting as exemplified in Figure 3B. Next, to determine PMCA transport activity for Annexin V+ SLHC and CD71+ RBCs from the L-fraction, we monitored changes in Fluo-4 fluorescence intensity over time after stimulation with Yoda1, and the rate of Ca^2+^ efflux was measured as changes in Fluo-4 fluorescence intensity in arbitrary units (A.U.) per minute. We observed that the Ca^2+^ level increased in both SLHC as well as in CD71+ RBCs (robust changes in this case) after Yoda1 supplementation (Figure 3C). Compared to CD71+ cells, the changes in Ca^2+^ level over time were significantly lower in Annexin V+ SLHC cells (−0.45 ± 0.063 A.U./min vs −0.07 ± 0.02 A.U./min), indicating inhibited PMCA transport activity in these cells. 

In the next series of experiments, we tested the involvement of Piezo1 in Ca^2+^ transportation in SLHC cells. First, by using hypoosmotic stress as a stimulus, we compared Piezo1-mediated Ca^2+^ transport between CD71+ cells and SLHC cells of the L-fraction. Exposure of RBCs to acute hypoosmotic shock by reducing the extracellular osmolarity from 330 to 220 mOsm caused swelling, which we monitored as a change in forward scatter in both CD71+ and in SLHC cells along with Ca^2+^-dependent fluorescence (Appendix A). We did not find any detectable changes in [Ca^2+^]_i_ in CD71+ cells when measured over 1–1.5 min after the stretch was applied (Figure 3D). However, in SLHC cells, we found some minor, but discernible increase in Ca^2+^ levels above the basal one (Figure 3D). Further experiments were performed in the presence or absence of Yoda1 and/or GsMTx4 in SLHC cells suspended in media with or without Ca^2+^. Supplementation of the extracellular Ca^2+^ increased [Ca^2+^]_i_ levels in SLHC cells (Figure 3E). Notably, in the previous section, no changes in Na^+^-dependent fluorescence was observed with extracellular Ca^2+^ supplementation (Figure 2D). Treatment of SLHC RBCs with Yoda1 in the presence of extracellular Ca^2+^ resulted in an increase in Ca^2+^-dependent Fluo-4 fluorescence intensity, while upon inhibition of Piezo1 with GsMTx4, we found reduced Fluo-4 fluorescence (Figure 3E). Further comparative analysis showed that while inhibition of Piezo1 with GsMTx-4 significantly decreased Fluo-4 fluorescence intensity in SLHC cells, similar treatment did not produce any significant changes in [Ca^2+^]_i_ in CD71+ L-fraction cells (Figure 3F), indicating a hyperactive Piezo1 in SLHC cells. Nevertheless, from this set of data, we can conclude that Piezo1 plays a prominent role in the overloading of Ca^2+^ as well as Na^+^ in SLHC cells. However, reviewing the changes in [Ca^2+^]_i_ as well as [Na^+^]_i_ upon Piezo1 inhibition (Figure 2 and Figure 3), we cannot exclude the possibility of the involvement of other nonselective ion channels in Ca^2+^ regulation in the SLHC cells. 

In the above section, we revealed that in SLHC RBCs showing suppression of PMCA and Ca^2+^ overload, Piezo1 transports a significant amount of Na^+^. We have modeled these settings to investigate whether Piezo1 is capable of transporting Na^+^ in the cells presented with inhibition of PMCA and Ca^2+^ overload. We treated non-fractionated RBCs with Ouabain or with Na-orthovanadate in the presence of extracellular Ca^2+^ to mimic inhibition of NKA, or both NKA and PMCA, respectively, and measured [Ca^2+^]_i_ and [Na^+^]_i_. As follows from Figure 4A, orthovanadate(vanadate)-treated cells were overloaded with Ca^2+^ compared to non-treated or Ouabain-treated cells. Piezo1 stimulation with Yoda1 further enhanced [Ca^2+^]_i_ in all these conditions to the same extent in the cells with or without inhibition of the ATPases (Figure 4B). However, this was not the case for the changes in [Na^+^]_i_. While inhibition of NKA did not cause any difference in [Na^+^]_i_ compared to non-treated control when both were exposed to Yoda1, treatment of the cells with both NKA and PMCA inhibitor resulted in [Na^+^]_i_ as well as Ca^2+^ overload in the presence of Yoda1 (Figure 4C), suggesting Piezo1 mediated Na^+^ transport only in Ca^2+^-overloaded cells in this case.

## 4. Discussion

In the present study, we have identified a small fraction of RBCs with senescent-like features that undergo density reversal. We found these cells in low-density fraction along with reticulocytes while performing fractionation in Percoll density gradient. Importantly, we found Piezo1 was one of the candidate proteins involved in terminal swelling of these cells. We have provided evidence regarding Piezo1-mediated Ca^2+^-dependent Na^+^ influx as one of the probable mechanisms causing the small proportion of senescent RBCs to swell. This striking observation is somewhat counterintuitive, as Na^+^ permeability of Piezo1 is low in the presence of the divalent cations, in particular, in the presence of Ca^2+^, which is expected to competitively inhibit Na^+^ uptake by Piezo1. Higher selectivity of Piezo1 for Ca^2+^ compared to that for Na^+^ and K^+^ was demonstrated earlier [21]. If this were the case for Piezo1 and other nonselective cation channels, the presence of 1.8–2 mM of Ca^2+^ in plasma would turn these channels into relatively selective Ca^2+^-transporting ion pathways in SCHC cells. However, uncompensated Ca^2+^ overload accompanied by the inactivation of PMCA in these cells decreased the preference of the Piezo1 channels for Ca^2+^ over Na^+^. Therefore, the findings of the present study support the following hypothetic molecular mechanism of TDR covering Ca^2+^ overload as well as high Na^+^ despite having fully active NKA in SLHC RBCs as shown in the scheme in Figure 5. We suggest that TDR is initiated by chronic Ca^2+^ overload originating most likely from a combination of overactivation of Piezo1 (and nonselective cation channels as well) and the inactivation of PMCA. An increase in Na^+^ leak through the nonselective cation channels is promoted by PMCA inactivation and Ca^2+^ overload. Overloading of Na^+^ (through Piezo1 as well as other transporters) cannot be compensated by NKA leading to the loss of transmembrane Na^+^ gradient and swelling of SLHC RBCs (Figure 5).

Indeed, this study was performed ex vivo using fractionated RBCs, which pose certain limitations to our conclusions. The settings we have used allowed us to concentrate on cells of interest that contribute to less than a percent of the total RBC population in the blood of healthy humans. However, we could only observe a “snapshot” in time and could not follow the fate of SLHC RBCs in the body. As we did not use any cell labeling suitable for direct detection of RBC age by flow cytometry (e.g., biotinylation and re-introduction of BRC into the circulation), we could not confirm that the SLHC RBC subpopulation was truly senescent. Finally, a comparison of the abundance and properties of SLHC RBCs in wild-type mice and those deficient for Piezo1 channel in myeloid lineage as well as similar studies in the blood of patients with Piezo1 gain-of-function and loss-of-function mutations (xerocytosis) may help to unravel the possible role of this ion channel in RBC clearance in the future.

During RBCs’ senescence, the changes in the abundance and activities of membrane proteins such as the ion channels and pumps along with the alterations in the ionic contents of the cells are well-known [10,22,23] and supported by our findings (Figs. 2–4). Increased [Ca^2+^]_i_ is believed to be one of the markers for terminal RBCs’ senescence. High [Ca^2+^]_i_ causes activation of scramblase and exposure of PS to the outer leaflet of the cells. In this scenario, Lang et al. proposed that high [Ca^2+^]_i_ causes shrinkage of RBCs, which marks the event termed as “Eryptosis” [24]. Recently, an interesting study from Klei et al. [25] showed that human spleen red pulp macrophages (RPMs) are filled with RBC ghosts that are formed due to prolonged retention of terminally swollen senescent RBCs under low shear stresses. Subsequently, these lysed RBC ghosts were suggested to be cleared by macrophages. Unlike Eryptosis, in TDR, the senescent cells swell, raising the possibility for these cells to undergo intravascular hemolysis and their splenic retention and clearance as suggested by Klei et al. [25]. Our findings, therefore, call for further investigations to elucidate the role of nonselective ion channels and PMCA in splenic retention and clearance.

## Figures and Tables

**Figure 1 cells-13-01363-f001:**
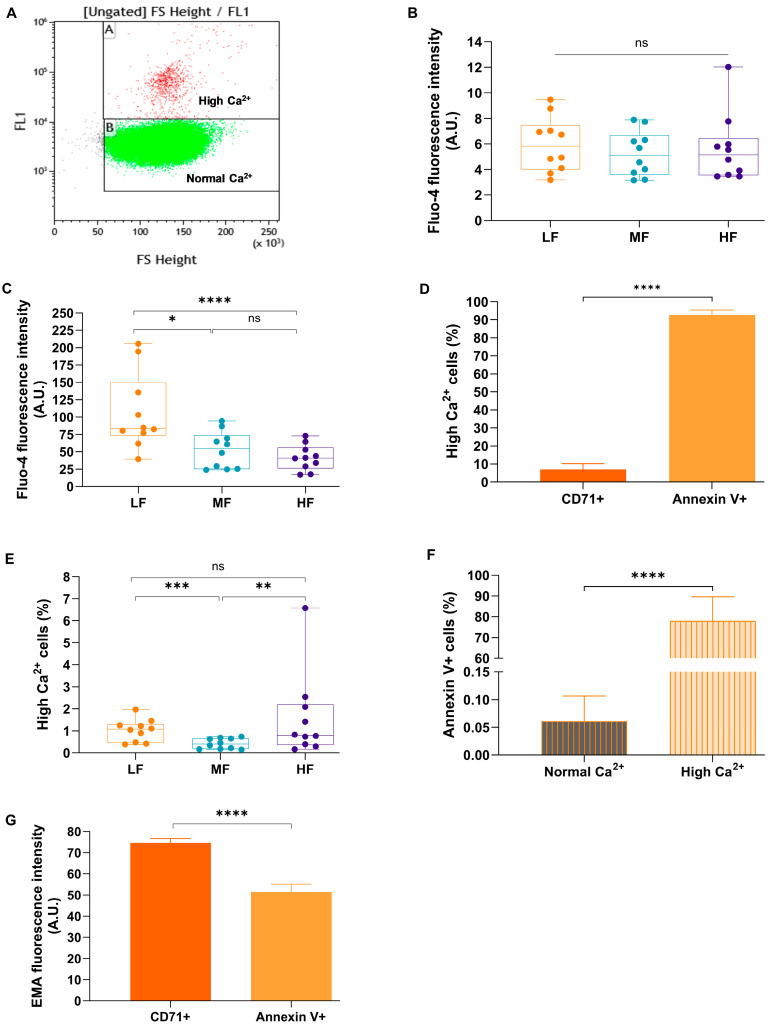
Identification of low-density senescent cells in the “High Ca^2+^” cell population. (**A**) An example of gating of normal and high Ca^2+^ cells based on their fluorescence intensity in unfractionated RBCs assessed using flow cytometry. Fluo-4 fluorescence intensity in cells gated as “Normal Ca^2+^” (**B**) (*n* = 10) and “High Ca^2+^” (**C**) cells (*n* = 10); (**D**) the percentage of “High Ca^2+^” gated cells (*n* = 10) which are significantly different among different fractions (Student’s paired *t*-test); (**E**) % of high Ca^2+^ cells in Annexin V+ gate (*n* = 8) and (**F**) % of Annexin V+ cells in “Normal Ca^2+^” and “High Ca^2+^” cell population (*n* = 29) showed a significant difference (paired Wilcoxon test); (**G**) Band 3 abundance was measured with EMA-fluorescence in the L-fraction of CD71+ and Annexin V+ cells (*n* = 4, paired Wilcoxon test). LF: L-fraction, MF: M-fraction, HF: H-fraction, data in (**B**–**G**) are presented as mean ± SD. * *p* < 0.05, ** *p* < 0.01, *** *p* < 0.001 **** *p* < 0.0001.

**Figure 2 cells-13-01363-f002:**
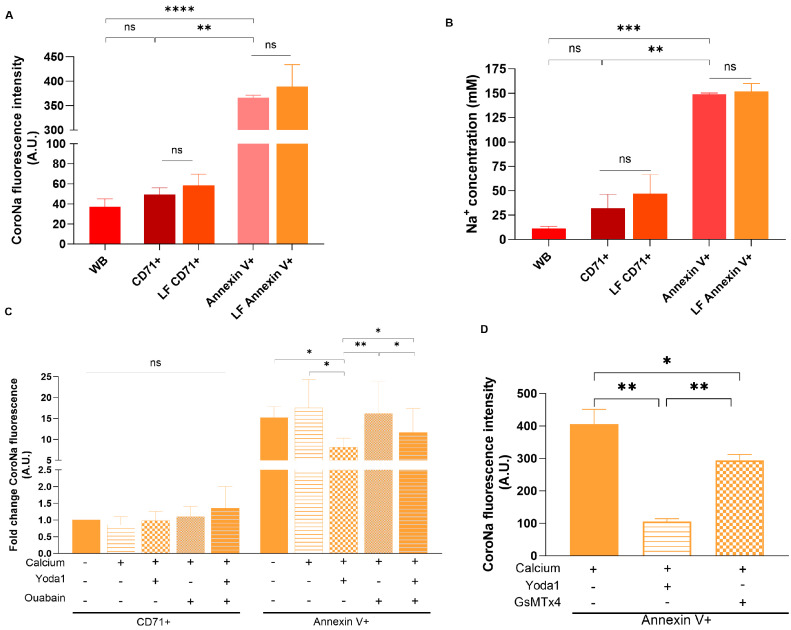
High intracellular Na^+^ and its pharmacological modulation in Annexin V+ SLHC cells. (**A**) CoroNa Green-AM fluorescence intensity in unfractionated whole blood (WB) and in CD71 and Annexin V+ RBCs (*n* = 3); (**B**) measurement of [Na^+^]_i_ in unfractionated whole blood (WB) and in CD71 and Annexin V+ RBCs (*n* = 3); (**C**) changes in CoroNa Green-AM fluorescence intensity when CD71+ or Annexin V+ cells were treated in presence/absence of extracellular Ca^2+^ (1.8–2.0 mM) and incubated with Yoda1 (1.5 µM) and/or Ouabain (100 µM) (*n* = 3); (**D**) changes in CoroNa Green-AM fluorescence intensity in Annexin V+ cells upon treatment with Yoda (1.5 µM) or with GsMTx4 (2.5 µM) (*n* = 3). All data are presented as means ± SD and the significance was tested with Student’s paired *t*-test. * *p* < 0.05, ** *p* < 0.01, *** *p* < 0.001 **** *p* < 0.0001.

**Figure 3 cells-13-01363-f003:**
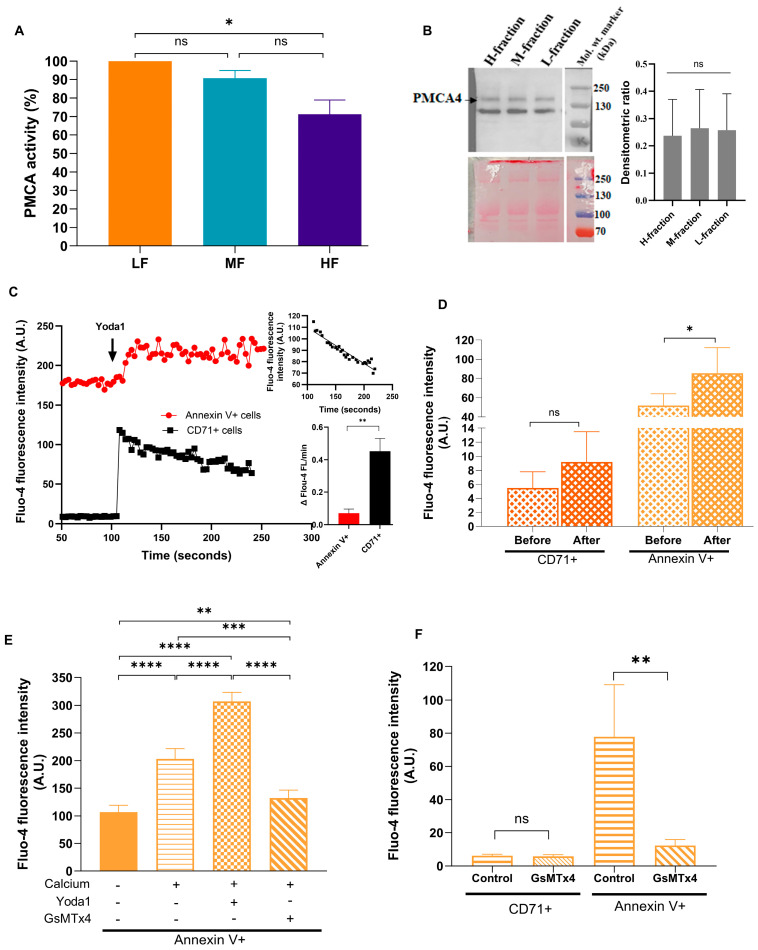
Intracellular Ca^2+^ level and its pharmacological modulation in RBCs. (**A**) PMCA activity in M- and H-fraction relative to their L-fraction measured with spectrophotometry (*n* = 3, paired Wilcoxon test). (**B**) PMCA4 abundance in L-, M-, and H-fraction was measured with Western blot (*n* = 3) (PMCA4 antibody: JA9, abcam ab2783). (**C**) Changes in Ca^2+^ fluorescence intensity over time in CD71+ as well as in Annexin V+ cells after stimulation with Yoda1 (*n* = 3). The inset images show how we calculate the rate of Ca^2+^ efflux (the slope of the curve) via PMCA. (**D**) Changes in [Ca^2+^]_i_ measured with Fluo-4 AM before and after hypoosmotic stress test (*n* = 5). (**E**) Modulation of [Ca^2+^]_i_ in Annexin V+ SLHC cells upon treatment with Yoda1/GsMTx4 (*n* = 6). (**F**) Comparative analysis of Fluo-4 fluorescence intensity in CD71+ vs Annexin V+ cells upon Piezo1 inhibition (*n* = 3). LF: L-fraction, MF: M-fraction, HF: H-fraction, PMCA: Plasma membrane Ca^2+^-ATPase. Data are presented as mean ± SD. * *p* < 0.05, ** *p* < 0.01, *** *p* < 0.001, **** *p* < 0.0001.

**Figure 4 cells-13-01363-f004:**
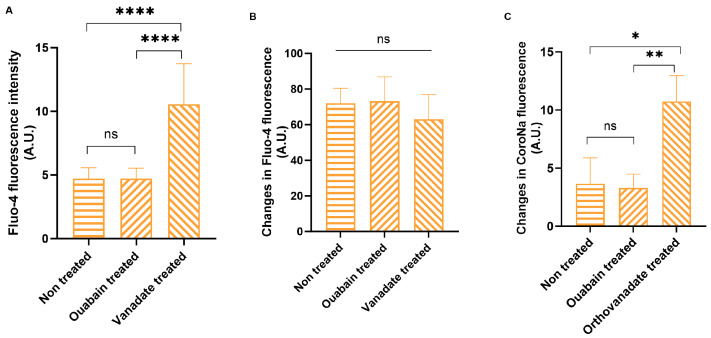
Piezo1 transports Na^+^ in Ca^2+^-loaded cells. (**A**) Measurement of basal Ca^2+^ as well as (**B**) changes in [Ca^2+^]_i_ in whole RBCs upon Ouabain and Na-orthovanadate treatment. (**C**) Changes in intracellular Na^+^ in whole RBCs upon Ouabain and Na-orthovanadate treatment. Data are presented as mean ± SD (*n* = 3, Student’s paired *t*-test), * *p* < 0.05, ** *p* < 0.01, **** *p* < 0.0001.

**Figure 5 cells-13-01363-f005:**
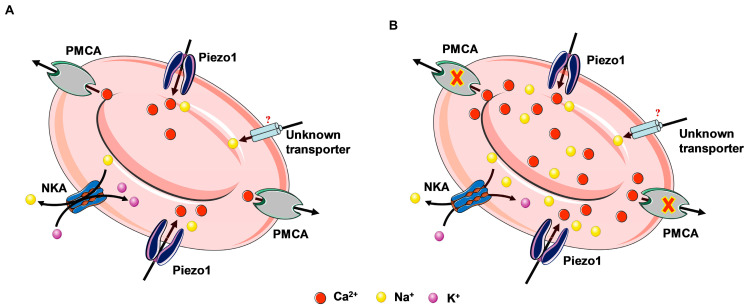
Proposed hypothesis. Hyperactivation of Piezo1 and inhibited PMCA activity leads to Ca^2+^ as well as Na^+^ overload in L-fraction Annexin V+ SLHC cells (**B**) compared to CD71+ cells (**A**).

## Data Availability

The raw dataset may be found on the Zenodo platform (DOI:10.5281/zenodo.5839743).

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
