# Peer review of "Role of Piezo1 in Terminal Density Reversal of Red Blood Cells"

_cells, 2024, doi:10.3390/cells13161363_

Round 1

Reviewer 1 Report

Comments and Suggestions for Authors

Dey et al. from the Bogdanova lab describe the role of Piezo1 in terminal density reversal of red blood cells (RBCs). The study comes from a group of world leading experts in the field and represents a timely addition to the literature. The authors used Percoll density  gradients to separate Low, Medium and High density RBC fractions (young, mature and senescent, respectively). They then focus on calcium levels in each fraction via flow-cytometry (Fluo-4 signal). While they did not observe differences in the general population, in the high Ca2+ sub-fraction they noticed the presence of some younger cells as well. However, this population is relatively small (0.5-2%) and perhaps biologically negligible. Yet, this observation may be relevant to the community and, in my opinion, warrants dissemination – at the very least to note that Percoll density gradients-based separation of young and old RBCs is an imperfect (yet still overall reliable) approach.

The study holds some limitations in that no mechanistic manipulation of Piezo1 in tractable systems was performed (e.g., Piezo1 KO mice, individuals with hereditary xerocytosis carrying polymorphisms in the region coding for Piezo1), nor orthogonal strategies to track RBC ages in vivo were performed (e.g., biotinylation). However, the authors did use Yoda1 to modulate Piezo1 activity. At the net of these considerations, which should be discussed briefly somewhere in a paragraph about “limitations” at the end of the discussion section, the paper does represent a solid contribution to the literature. As such, I am glad to recommend the publication of this manuscript upon (very) minor revisions (i.e. adding considerations on limitations noted above).

Author Response

Comment: 

The study holds some limitations in that no mechanistic manipulation of Piezo1 in tractable systems was performed (e.g., Piezo1 KO mice, individuals with hereditary xerocytosis carrying polymorphisms in the region coding for Piezo1), nor orthogonal strategies to track RBC ages in vivo were performed (e.g., biotinylation). However, the authors did use Yoda1 to modulate Piezo1 activity. At the net of these considerations, which should be discussed briefly somewhere in a paragraph about “limitations” at the end of the discussion section, the paper does represent a solid contribution to the literature. As such, I am glad to recommend the publication of this manuscript upon (very) minor revisions (i.e. adding considerations on limitations noted above).

Response: We thank the reviewer for the constructive criticism of our manuscript. Revised version of the text (lines 465-476) contain the paragraph in which limitations of the approach we used is discussed as suggested by the reviewer. 

"

Indeed, this study was performed ex vivo using fractionated RBCs, which pose certain limitations to our conclusions. The settings we have used, allowed us to concentrate cells of interest that contribute to less than a percent of the total RBC population in the blood of healthy humans. However, we could only observe a “snapshot” in time and could not follow the fate of SLHC RBCs in the body. As we did not use any cell labeling suitable for direct detection of RBC age by flow cytometry (e.g. biotinylation and re-introduction of BRC into the circulation), we could not confirm that the SLHC RBC sub-population was truly senescent. Finally, a comparison of the abundance and properties of SLHC RBCs in wild-type mice and those deficient for Piezo1 channel in myeloid lineage as well as similar studies in blood of patients with Piezo1 gain-of-function and loss-of-function mutations (xerocytosis) may help to unravel the possible role of this ion channel in RBC clearance in the future."

Reviewer 2 Report

Comments and Suggestions for Authors

The manuscript by Dey et al. entitled “Role of Piezo1 in terminal density reversal of red blood cells” presents an interesting new concept about the factors leading to the formation of “senescent-like high calcium” type of red blood cells, i.e. senescent cells with preserved hydration and density. The phenomenon was first reported more than 20 years ago but its nature remained elusive. The authors of the current manuscript identify specific ion transporters/channels that contribute to the abnormal Ca2+ and Na+ ion uptake in those cells. Thus, the work unveils new insights into the mechanisms of ion regulation in red blood cells which makes it innovative and suitable for publication in Cells. The paper is clear and concise. The methods are properly presented. The experimental work is properly described and discussed. However, there are some minor issues that need revision in my opinion:

In the abstract the authors use the term “loss of membrane”. Could you clarify what exactly this means, maybe lower cell volume?

Fig. 1A – it is not clear if the data presented are related to unfractionated RBC or to specific RBC fraction. Please clarity.

Caption of Fig. 1D – I recommend to change the “number of High Ca gated cells” with “percentage” or “relative abundance”.

In paragraph 3.2. the authors refer to Fig. S1 instead of Fig. S2.

Fig. 3 – panels G and H are missing in the figure but mentioned in the caption. Please revise caption and/or figure, and relevant text in the ms.

Fig. S1 – panels A and B are reversed.

Author Response

We thank the reviewer for careful and attentive reading of the manuscript! All his suggestions were accepted with gratitude.

Comment1: In the abstract the authors use the term “loss of membrane”. Could you clarify what exactly this means, maybe lower cell volume?

Response1: The exact interpretation of the findings in Fig 1G is a reduction in band 3 protein abundance, which results from the vesiculation and loss of membrane area. Cell volume is a function of cell membrane and hydration state, and, hence, does not precisely describe the process. We thus corrected the abstract text replacing the “loss of membrane” with “loss of membrane surface area associated with reduction in band 3 protein abundance”.

Comment 2: Fig. 1A – it is not clear if the data presented are related to unfractionated RBC or to specific RBC fraction. Please clarity.

Response 2: Indeed, this information was missing and is added to the revised version of the manuscript.

Comment 3: Caption of Fig. 1D – I recommend to change the “number of High Ca gated cells” with “percentage” or “relative abundance”.

Response 3:Thank you for this comment, we have changed the text accordingly.

Comment 4: In paragraph 3.2. the authors refer to Fig. S1 instead of Fig. S2.

Response 4:Thank you, this is corrected in a new version of the manuscript.

Comment 5: Fig. 3 – panels G and H are missing in the figure but are mentioned in the caption. Please revise caption and/or figure, and relevant text in the ms.

Response 5: Thank you for noticing it. We removed the G and H from the figure legend.

Comment 6: Fig. S1 – panels A and B are reversed.

Response 6: It is reversed, indeed, and is corrected now.